# Pool-Boiling Performance on Thin Metal Foils with Graphene-Oxide-Nanoflake Deposit

**DOI:** 10.3390/nano12162772

**Published:** 2022-08-12

**Authors:** Tadej Bregar, Matevž Vodopivec, Tim Pečnik, Matevž Zupančič, Iztok Golobič

**Affiliations:** Faculty of Mechanical Engineering, University of Ljubljana, Aškerčeva 6, 1000 Ljubljana, Slovenia

**Keywords:** graphene oxide, nanocoating, nucleate boiling, local heat flux, nucleation site density

## Abstract

The pool-boiling performance of water on thin metal foils with graphene-oxide deposition was studied. The boiling performance was evaluated both on fully coated surfaces, achieved by spin-coating, and surfaces with a laser-textured nucleation site, into which graphene oxide was added via drop-casting. During the experiments, a high-speed IR camera was used to obtain the transient temperature and heat-flux distribution. At the same time, a high-speed video camera was used to acquire synchronized bubble-growth recordings. In addition, a surface-wettability analysis was conducted for all the samples. In the case of fully coated samples, graphene-oxide deposition resulted in an increased number of active nucleation sites and an increase in the nucleation temperature, leading to a lowered nucleation frequency. Meanwhile, samples with a single laser-textured nucleation site enabled the analysis of isolated vapor bubbles, confirming that graphene-oxide deposition leads to a higher nucleation temperature, consequently resulting in a larger bubble-departure diameter and longer growth time. Two explanations for the results are proposed: the wettability of graphene-oxide deposition and the filling of surface microcavities with graphene-oxide nanoflakes.

## 1. Introduction

The process of pool boiling is a highly effective approach to heat transfer, which is utilized in different applications, such as power plants, nuclear reactors, heat exchangers, and other thermal-management equipment. Its intrinsic capacity to transfer heat by natural convection, allied with the forced convection inducted by the detaching bubbles and the contribution of latent heat removal, makes boiling a more efficient means of heat dissipation in liquid–surface interfaces, when compared to single-phase heat transfer [1]. The performance of nucleate boiling is strongly related to the boiling surface properties, type of working fluid, operating conditions, and supplied heat flux. Moreover, the variety of surface factors, such as roughness, topology, porosity, wettability, and wickability [2,3,4], might vary in time due to surface aging, fouling, oxidation or contamination [2,5,6,7]. Due to a plethora of influential parameters, the experimental approach is still the most reliable way to characterize the boiling performance on newly developed surfaces.

Many boiling-enhancement research studies to date have focused on boosting nucleate-boiling heat transfer by increasing the effective surface area, nucleation site density and distribution, and bubble-departure frequency, while decreasing the bubble-nucleation temperature and bubble wait time [8]. These enhancements can be achieved via active techniques such as liquid vibrations, surface rotation, or the use of high-intensity electric fields [9]. However, passive techniques via surface engineering have attracted the most attention in boiling enhancement due to their cost-effectiveness and the possibility of implementing such surfaces into different environments [10]. In practice, enhanced boiling surfaces can be constructed through additive, subtractive, or compound manufacturing methods, which are encompassed in nanoparticle deposition [11,12,13,14], micro and nanoporous structure preparation [15,16], the preparation of nanowire in nanotube structures [3,17], laser texturing [18,19], electrodeposition [20], and chemical vapor deposition [21]. It is important to note that many additive techniques capable of modifying surface wettability, wickability, and porosity can significantly increase the thermal resistance of the heated surface. This increases the temperature gradient along the heater thickness, reduces the heat-transfer performance, and also prevents the observation of local phenomena (i.e., transient-temperature fields) when boiling is performed on thin substrates or samples that are transparent to infrared (IR) light. The development of ultra-thin or highly thermally conductive layers is therefore very desirable, both for boiling research and for final intended heat-transfer applications. One solution that combines both desired characteristics is the utilization of graphene and graphene oxide (GO), which is also investigated in this paper.

Until now, several investigations have been published on research on pool-boiling enhancement with graphene oxide, reduced graphene oxide (rGO), graphene nanoplatelets (GNPs), or carbon nanotubes (CNTs). Lay et al. [22] used GNP-coated surfaces to enhance the nucleate pool boiling of water through changes to the wettability of the surface and the fast water-permeation property of a thermally cured nanoporous GNP structure. Cured GNP coating enhanced the heat-transfer coefficient (HTC) by 151% and the vapor mass flow rate by 154%. A comparison between the pool-boiling heat transfer of GNP-, functionalized-GNP-, and CNT-based nanofluids was conducted by Akbari et al. [23]. The authors found that in all cases, the critical heat flux (CHF) and HTC increased compared to the DI water results, with functionalized GNPs exhibiting the largest CHF enhancement, of 72%. Ahn et al. [24] studied the effect of a rGO coating on CHF enhancement by depositing the coating through the boiling of rGO colloid. Thin-wire heaters were used, and since rGO flakes were negatively charged after synthesis, the coating was biased on the anode side of the wire heater. Two-sided coating provided uniform rGO coating, which resulted in a maximum CHF enhancement of 320%. Similar experiments with a wire heater were conducted by Kim et al. [25], the difference being in their use of a GO colloid rather than rGO. The CHF enhancement of 63% compared to distilled water was attributed to the heat-dissipation effects of the GO layer. Research by Seo et al. [26] compared the pool-boiling CHF enhancement of nonporous graphene and porous GO deposition with highly wettable FC-72 as the working fluid. The nonporous layer, due to its thermal properties, provides a 9% CHF increase compared to the bare surface, while the porous GO provides a 90% increase due to its surface morphology and porous structure. Sezer et al. [27] investigated the pool-boiling enhancement of hybrid GO/CNT coatings, which resulted in greater CHF and HTC enhancements (maximum of 153% and 100% for CHF and HTC, respectively) compared to enhancement by GO-only coating (maximum 124% and 90% for CHF and HTC, respectively). Khan et al. [28] investigated the boiling performance of GO nanofluid on microporous surfaces to obtain a hybrid surface. Separate microporous surface experiments and GO nanofluid experiments on a reference surface showed an increase in performance (maximum 231% CHF increase for GO and maximum 347% average HTC increase for microporous surface), while the hybrid surface severely decreased the boiling performance in terms of HTC and CHF.

Despite all these efforts, there are still no published data about the influence of GO and rGO coatings on the key boiling parameters, such as the nucleation-site density and distribution of nucleation sites, bubble frequency, growth time and wait time, bubble-footprint radius, transient-temperature fields, and local heat fluxes. Moreover, little research has been conducted that combines techniques of already researched surface structuring at the micro level with the addition of GO/rGO coatings that can provide adequate surface nanostructures for improving boiling process.

To fill this knowledge gap, saturated pool-boiling experiments were conducted on GO-coated thin metal foils with and without laser-textured areas (i.e., predefined nucleation sites, using a similar approach to that described in [29,30]), to obtain insight into the effect of GO on the boiling process. A high-speed IR camera was used to measure the transient-temperature fields on the bottom side of thin stainless-steel foil, while, simultaneously, a high-speed video camera captured the growth of the bubbles on top of the foil. With the use of a custom-made data-processing algorithm, the locations of the active nucleation sites and the characteristics of the bubble growth were determined. This research is a step towards a better understanding of boiling phenomena on thin substrates with graphene-oxide deposition and towards the development of boiling surfaces to be used for isolated bubble analysis.

## 2. Materials and Methods

### 2.1. Pool-Boiling Setup and Measurement Procedure

The pool-boiling setup, schematically presented in Figure 1, encompassed a boiling chamber measuring 50 × 50 × 50 mm^3^, a high-speed IR camera (SC6000; FLIR Systems, Wilsonville, OR, USA), a high-speed video camera (FASTCAM UX100; Photron, Tokyo, Japan), a light source (LED LT2BC048036-W; Opto Engineering, Mantova, Italy), and a data control and acquisition system based on CompactRIO real-time controller. The chosen boiling chamber by design allowed viewpoints from all four sides, with the heater unit mounted at the bottom of the chamber, consisting of a ceramic base and two electrical contacts, designed to both hold and power the stainless-steel foil measuring 15 × 14 mm^2^. The ceramic base included a hole measuring 10 × 12 mm^2^ in its center to allow IR-camera recordings of transient-temperature fields on the bottom of the foil. At the same time, synchronized high-speed video recordings of the boiling process were made through the side window of the boiling chamber. To enable IR thermography, the bottom side of each foil was painted using a custom-made high-emissivity paint. The foils used in the experiments were 25-micrometer-thick stainless-steel foils and were heated using the Joule effect through DC power supply.

The working fluid used was degassed distilled water. In order to ensure both complete degassing and constant saturation conditions (100 °C at 1 atm) during the experiment, an additional cartridge heater was inserted at the bottom of the boiling chamber to preheat and degas the working fluid. For maintaining a constant water level during boiling, a reflux condenser was attached to the top of the boiling chamber.

The IR and high-speed video cameras were set to record at 2000 Hz. An external trigger was used to activate the high-speed camera, which in turn activated the IR camera through a logic circuit to obtain synchronized recordings. The data-acquisition system, which measured the bulk temperature of the working fluid and the electrical heating power, was activated manually at the same time as the heater unit. The bulk temperature was measured with thermocouples inserted in the boiling chamber. A reference resistor was used to calculate the current applied to the foils through module NI-9211 (M1), while module NI-9205 (M2) measured the voltage drop across the foil. Both values, along with the effective area of the foil, were needed to calculate the heat flux applied to the foil, which is further explained in Section 2.3. Module NI-9263 (M3) could be used to control the power supply via PC, but this option was not used in our experiments.

The framerate and position of the IR camera allowed a spatial resolution of 115 µm pixel^−1^ and high-speed video camera had a resolution of 14 µm pixel^−1^. The expanded relative uncertainty of the average heat flux due to Joule heating was estimated to be ~0.5%, and the absolute measurement uncertainty of the temperature obtained with the IR camera was 1 K.

Measurements were performed at three different heat -flux levels, namely 50 kW/m^2^, 100 kW/m^2^, and 150 kW/m^2^. It is known that due to limited heat diffusion inside the thin foils [31,32], the burnout point for 25-micrometer stainless steel heater is reached in the range of about 0.2–0.6 of the hydrodynamic limit (1.2 MW/m^2^ for saturated pool boiling of water). Relatively low heat-flux levels were thus chosen to avoid reaching the burnout point (and to prevent any undesired surface-morphology and chemistry changes due to high-temperature effects), as well as to allow investigation of single bubble growth on the laser-textured nucleation side without triggering any parasitic boiling.

### 2.2. Graphene-Oxide-Nanoflake Deposition

Graphene oxide (GO)-nanoflake deposition was performed using GO dispersion with a mass concentration of 0.25%. Two different methods of deposition were utilized.

Spin-coating deposition technique was used to fully coat the effective area of the foils. GO dispersion volume of 0.25 mL was spread across each foil and spin-coated for 60 s at 2000 rpm. The spin-coating process was repeated three times for a total of three deposited layers on each foil. Each coated foil was then left to airdry at room conditions overnight. In order to ensure a stable coating, further thermal annealing of all foils was performed at two different temperatures, 130 °C and 250 °C, with samples named GO1 and GO2, respectively. The annealing process lasted 1 h. As previously shown by other researchers, thermal annealing causes a volume reduction in GO film [33,34]. One of the goals of this work was also to determine how different annealing temperatures affect the reduction process (and, consequently, wettability of the surfaces) and how this, in turn, affects boiling performance. A bare stainless-steel sample, named REF, was used in the experiments as a reference surface.

Second method of GO deposition was drop casting, which was used in combination with laser texturing of nucleation sites. Laser texturing is an efficient method to create surface microcavities, which have already been proven to act as stable active nucleation sites in pool-boiling experiments [29,35,36]. Each nucleation site was made with laser ablation using a pulsed fiber laser (λ = 1064 nm) with a scanning head and motorized platform for setting the vertical position of F-theta lens (JPT Opto-electronics Co., Ltd., Shenzhen, China, YDFLP-E-30-M7-S-R). Laser was set to average power of 3 W, pulse duration of 60 ns (duration at 10% of the peak power), and pulse frequency of 100 kHz. The resulting pulse fluence was 6.1 J/cm^2^. Laser beam with a diameter of d = 25 µm was then guided by the scanning head in straight parallel lines across an area measuring 300 × 300 µm^2^ with scanning velocity of 450 mm/s. The distance between the lines was increased by steps of 5 µm from 15 to 35 µm and then decreased back to 15 µm.

Next step involved a drop of 0.25% GO dispersion, with a volume of 2 µL drop-casted onto the laser-textured nucleation site of each foil. A hot-air gun was used to accelerate the evaporation of the liquid so that a 1-mm spot of GO remained on the surface atop the nucleation site. The foil was then thermally annealed at 130 °C for 1 h to obtain a stable spot of GO. This surface with added GO was named NS-GO1, while the reference surface with a bare laser-textured nucleation site was named NS-REF.

To identify the effect that GO film and its reduction had on wettability and roughness of the surface, contact-angle measurements (Ossila Contact Angle Goniometer, Ossila Ltd., Sheffield, UK) and profilometry (SJ-301; Mitutoyo Corporation, Kanagawa, Japan) were carried out. The contact-angle measurements were obtained on bare stainless-steel reference surfaces, after spin-coating procedure and thermal annealing at the two different temperatures. Results are shown in Table 1 and Figure 2, respectively. On laser-textured nucleation site (i.e., see Figure 2a), the contact-angle measurements were not possible due to small size of the textured area, and only surface roughness was determined in this case.

As reported in Table 1, the reference surface was slightly hydrophobic, with a contact angle of 98°, while airdried graphene-oxide film changed the wettability to hydrophilic, with a contact angle of 29°. Thermal annealing caused reduction in GO film and contact angles increased to 48° and 77° for surfaces GO1 and GO2, respectively. Thermal reduction at higher temperatures increases the amount of oxygen-containing functional groups removed from GO film, which increases carbon-to-oxygen (C/O) ratio [37,38,39]. Higher carbon content (C/O ratio) causes lower surface wettability, as properties of reduced GO become similar to those of graphene, which is known to be hydrophobic [33,40]. This corresponds to larger contact angles compared to reduction at lower temperatures. Both fully coated surfaces, however, still exhibited lower contact angles compared to the reference bare surface. All non-textured samples (REF, GO1, and GO2) exhibited virtually the same surface roughness, since all the values were within the measurement uncertainty (~0.03 µm). This shows that graphene coating did not influence the roughness parameters significantly, which makes it suitable for independent wettability studies. With laser texturing, we were able to significantly increase surface roughness and also create microcavities, which are favorable to trapping of vapor and act as active nucleation sites. In this case, the graphene coating had a certain effect, as it partially filled the surface valleys. This was confirmed by SEM imaging on and also reduced peak-to-valley height (*Rz*) of the profile.

### 2.3. Data Reduction and Image Processing

In order to calculate time-dependent heat-flux distributions, certain assumptions were made. The stainless-steel foils were considered to have a constant thickness of 25 µm and adiabatic conditions on bottom side of the foil, which was in contact with hot air, were assumed. It was additionally assumed that the electrical resistance of the foil was uniform. Based on these, the average heat flux due to the Joule effect (${\dot{q}}_{el}$) was calculated as the ratio between electrical power dissipated across the stainless-steel foil and its effective boiling area of 14 × 15 mm^2^. Using the transient-temperature fields measured with the HSIR camera, $T_{w}\left(x,y,t\right)$, one can calculate the time-dependent heat-flux distribution with the following equation:(1)$${\dot{q}}_{loc}\left(x,y,t\right)={\dot{q}}_{el}-\delta \;\rho \;c_{p}\frac{∆T_{w}\left(x,y,t\right)}{∆t}+\delta \;k\left(\frac{{∆}^{2}T_{w}\left(x,y,t\right)}{∆x^{2}}+\frac{{∆}^{2}T_{w}\left(x,y,t\right)}{∆y^{2}}\right)$$
where Δ*t* equals 0.5 ms (corresponding to 2000 fps) and Δ*x* (=Δ*y*), i.e., the pixel size, equals 115 µm. Material properties of the foils, used in the equation, are as follows: specific heat capacity *c_p_* is 500 J/kgK, thermal conductivity *k* is 16.2 W/mK, and density ρ is 7500 kg/m^3^. The effect of high-emissivity paint was neglected in Equation (1) due to its low thickness (~5 µm) and high thermal conductivity (78.0 W/mK) [41].

With local heat-flux values calculated and stored in a three-dimensional matrix, it was possible to identify locations of active nucleation sites on boiling surfaces. Detection algorithm was based on equation developed by Ravichandran et al. [42], which detects the zero-crossing of second derivatives by *x* and *y* of the heat flux. Although the algorithm provided satisfactory results, the improved version of the algorithm by Pečnik et al. [43] was used in our final analysis. The modified algorithm also takes into account the temperature gradients and is based on the following equation:(2)$$p_{n}\left(x,y\right)=\overset{n_{f}-1}{\underset{f=1}{\sum }}\left(\frac{dq_{w}\left(x,y,f\right)}{dt}>0\right)\cdot \left(\frac{dT\left(x,y,f\right)}{dt}<0\right)d\cdot \;\left(\frac{d^{2}q_{w}\left(x,y,f\right)}{dt^{2}}>0\right)\cdot \left(\frac{d^{2}T\left(x,y,f\right)}{dt^{2}}<0\right)$$

Implementing Equation (2) over the transient local temperature and heat-flux data, the results were obtained and are presented in Figure 3 and Figure 4. The first row of the figure represents images, produced by the algorithm, with nucleation sites seen as circular areas. Second row adds circles that highlight nucleation sites in the existing pattern and determines their original center. Blue-color circles represent the nucleation sites that were correctly found by the algorithm and red color represents sites that were not detected and were added manually. Reference surface is shown in Figure 3 and surface GO2 is shown in Figure 4.

Based on detected locations of active nucleation sites, it was possible to determine the surface temperature just before the nucleation occurred, i.e., the nucleation temperature. Plot of the wall superheat (i.e., the temperature difference between the heater wall and the liquid boiling point) profile shows how the temperature rose until nucleation temperature was reached, dropped when the bubble formed during bubble growth (*t_g_*), and rose again once the bubble detached during the waiting period (*t_w_*). Figure 5 shows wall superheat profile for a single nucleation occurrence along with growth and wait times for the specific bubble. Growth and wait times are important for determining the nucleation frequency *f_nuc_*, which is calculated using Equation (3).
(3)$$f_{nuc}=\frac{1}{t_{g}+t_{w}}$$

## 3. Results and Discussion

### 3.1. Boiling Performance on Foils Fully Coated by Graphene-Oxide Nanoflakes

Using transient-temperature field measurements, the average wall superheat for each surface at the three heat fluxes used in the experiments was determined. The heat-transfer coefficient corresponding to the wall superheats and average local heat flux values were also calculated. The results are shown in Figure 6a,b. Average wall superheat values increased as the heat flux increased and relative differences between results were similar at each heat flux. Average wall superheat was the lowest on surface REF and it correspondingly achieved the highest heat-transfer coefficient. Since surface REF exhibited the highest contact angles, the wall superheat results were attributed to the higher hydrophobicity of this surface compared to surfaces GO1 and GO2. It has already been proven that hydrophobic surfaces require less energy for nucleation to occur [44]. Consequently, surface GO1, which exhibits the lowest contact angle, demonstrates the highest average wall superheat.

Considering the average wall superheat and HTC values, the GO film did not provide pool-boiling enhancement in terms of the average heat-transfer coefficient. However, the image-processing analysis showed that the GO film significantly increased the number of active nucleation sites, as can be seen in Figure 3 and Figure 4, and, at the same time, increased the nucleation temperature (Figure 6c). This effect was more noticeable at higher heat fluxes (100 and 150 kW/m^2^). At 50 kW/m^2^, nucleate boiling was not yet fully developed, which was confirmed by the lack of active nucleation sites and low nucleation frequency, shown in Figure 6d. The remaining results show that increased nucleation temperature and increased amount of active nucleation sites lower the nucleation frequency. On one hand, nucleation sites affect each other through coalescence and intensified hydrodynamics (which, in turn, affects the development of the thermal boundary layer), and, on the other hand, higher nucleation temperatures require more time for nucleation conditions to develop (during the waiting period), both of which result in lower nucleation frequency.

Similar to the results of the average wall superheat, the highest nucleation temperature occurred on surface GO1, with the lowest contact angles, while the opposite was observed for the surface REF. Correspondingly, the surface GO1 achieved the lowest, and the surface REF the highest nucleation frequencies. The surface GO2 generally fell between the other two surfaces for each determined boiling parameter (Figure 6). The change in surface wettability caused by GO deposition is recognized as an important factor that influences boiling parameters, as has already been shown in some other pool-boiling studies [45,46]. Additionally, GO nanoflakes can fill the natural microcavities of the surface, which can inhibit bubble formation; however, our results show that GO deposition provides enough cavities between nanoflakes, which increase the total number of active nucleation sites, which is also confirmed in Figure 6e.

### 3.2. Isolated Bubble Dynamics

Foils with a laser-textured nucleation site were used to analyze and compare isolated bubble dynamics between the bare nucleation site and a site with additional GO deposition. Figure 7 shows that the nucleation temperature on the surface NS-GO1 was significantly higher compared to the surface with the bare nucleation site. The nucleation frequency was, accordingly, much higher on the surface NS-REF, which corresponds to the larger number of nucleation events observed during the experiments.

Figure 8, Figure 9 and Figure 10 show the bubble-growth process of a single nucleating bubble for surfaces NS-REF and NS-GO1, respectively. The process was observed through synchronized images from high-speed video-camera recordings, transient temperature fields, local heat fluxes, and local heat-transfer coefficients. Additional videos of a single bubble growth can be found in Appendix A. It is shown that the surface NS-GO1 provided larger bubble-detachment diameter and that the bubble-growth period was longer (18.5 vs. 20 ms) compared to the NS-REF surface. The longer bubble-growth time and its larger diameter were consequences of the higher nucleation temperatures achieved by the surface NS-GO1, where more heat was transferred to the working fluid from the heating surface. The ineffectiveness of GO deposition in enhancing pool-boiling heat transfer (in terms of lowering nucleation temperature) was attributed to the wettability of the deposition and the combination of the surface modification by the laser texturing and the drop-casting of the GO dispersion. The laser texturing of the surface did create several microcavities that acted as possible nucleation sites; however, the addition of GO dispersion caused GO nanoflakes to fill the previously created microcavities, which inhibited their vapor-trapping potential and subsequent bubble nucleation. An SEM image and schematic of the GO nanoflakes filling the surface microcavities is shown in Figure 2c,i, as well as by the surface-roughness measurements reported in Table 1.

The temperature profiles underneath the active nucleation site are shown in Figure 8 for each tested heat flux. A clear difference in nucleation temperature and frequency can be seen, as well as a consistency in the nucleation temperature of each surface. This proves the steady-state conditions in the boiling chamber, which confirms the suitability of this experiment for future studies on isolated vapor bubbles.

In the last part of our analysis, presented in Figure 11, we compared the energy removed by a single bubble during the bubble-growth time. The thermal power due to the bubble evaporation for each timestep was computed through the integration of local heat flux underneath the bubble footprint. By integrating the thermal power over the entire bubble-growth period, it is possible to obtain the total energy removed by a single bubble due to liquid evaporation at the wall. It was clearly seen for all tested cases that the bubbles vaporizing from the graphene-coated nucleation site (NS-GO1) removed around 120 mJ of energy, which was significantly more than the bubbles on the bare sample (NS-REF), where the total evaporation energy increased from 40 mJ (at 50 kW/m^2^) up to 95 mJ (at 150 kW/m^2^). This was mainly attributed to the fact that hydrophilic-coated surfaces require higher nucleation temperatures and also produce larger bubbles compared to non-coated surfaces. Importantly, the bubble energy remained nearly constant at different heat fluxes for the NS-GO1 sample and increases in the heat flux mainly caused an increase in nucleation frequency (i.e., see Figure 7). On the other hand, the nucleation temperature and bubble departure diameter increased with the heat flux on the NS-REF surface, which also resulted in evaporation-energy increases. It can be concluded that the graphene-coated surfaces provided more stable bubble nucleation and larger nucleation-site density, but showed higher nucleation temperatures and a lower average heat-transfer coefficient compared to the non-coated surfaces.

## 4. Conclusions

In this work, the boiling process of water on thin metal foils with the application of graphene-oxide film was experimentally analyzed and compared with boiling on untreated foils. Foils with laser-textured nucleation sites were also manufactured, which allowed the analysis of the graphene oxide on isolated vapor bubbles. The transient-temperature fields on the foil were measured with a high-speed IR camera, while, at the same time, the process of the growth and detachment of the boiling bubbles on top of the foil was recorded with a high-speed video camera. Post-processing algorithms were used to analyze the transient-temperature fields and calculate the heat-flux distribution on the boiling surfaces and to determine the locations of the active nucleation sites.

The contact-angle measurements showed the higher hydrophobicity of the reference untreated surface compared to the surfaces with spin-coated GO. Additionally, higher annealing temperatures resulted in greater contact angles for the surface GO2 compared to the surface GO1. By analyzing the average surface superheat and corresponding heat-transfer coefficient, it was shown that the application of graphene oxide on the surface does not necessarily provide improvements to the heat-transfer coefficient during boiling. The reference surface with the highest contact angles exhibited the lowest average nucleation temperature, while the opposite was true for surface GO2. The wettability of the surface was recognized as a possible cause of the observed results. Importantly, the graphene-oxide film increased the number of active nucleation sites and, in combination with higher nucleation temperatures, resulted in decreased nucleation frequency.

High-speed and IR-camera recording showed that the laser-textured nucleation site allowed the observation of isolated vapor bubbles. The analysis of the transient-temperature fields confirmed that the GO deposition on the nucleation site inhibited bubble formation. The nucleation temperature of the surface with added GO was significantly higher, and the nucleation frequency was significantly lower than on the surface with the reference nucleation site. Consequently, the bubble on the nucleation site with GO provided larger a bubble-departure diameter and a longer growth/wait time. Two possible reasons for these results have been proposed: the wettability of graphene-oxide deposition and the phenomenon of the surface microcavities filling with GO nanoflakes due to a combination of laser texturing and GO coating. Furthermore, the bubble-departure diameter, nucleation temperature, and bubble-evaporation energy remained relatively constant on the graphene-coated samples in the 50–150 kW/m^2^ heat -flux range. From this point of view, graphene-coated surfaces provide more stable bubble nucleation.

Future studies may also focus on research into isolated vapor bubbles and investigate the influence of the size of a textured nucleation site on the formation of vapor bubbles. It would also be beneficial to investigate the influence of graphene-oxide films on the boiling of dielectric fluids, which are characterized by low surface energy and smaller nuclei. Graphene-oxide nanoflakes have the potential to create surface nanocavities, which may be used to create well-defined locations of active nucleation sites for low-surface-tension fluids.

## Figures and Tables

**Figure 1 nanomaterials-12-02772-f001:**
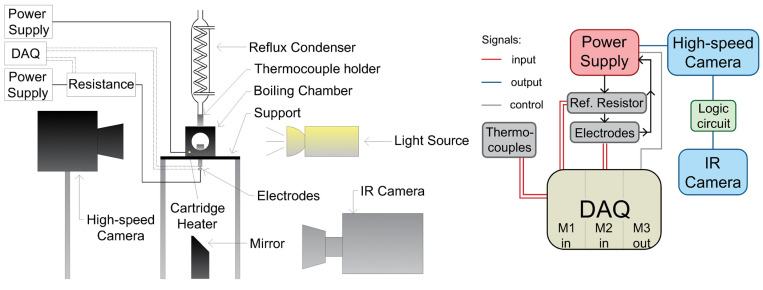
Schematic of the pool boiling setup (**left**) and a block diagram of the data-acquisition system (**right**).

**Figure 2 nanomaterials-12-02772-f002:**
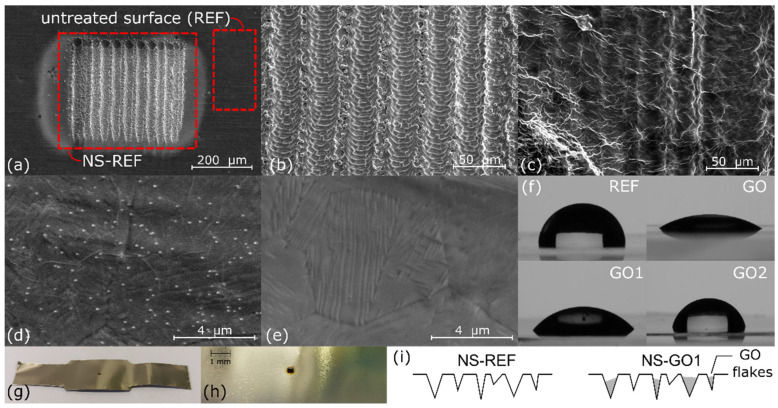
(**a**) SEM image of untreated REF surface and laser-textured nucleation site of NS-REF surface, (**b**) SEM image of NS-REF surface, (**c**) SEM image of NS-GO1 surface, (**d**) SEM image of surface GO1, (**e**) SEM image of surface GO2, (**f**) contact angles of DI water droplets on spin-coated surfaces, (**g**) experimental sample, (**h**) laser textured nucleation site, (**i**) schematics of GO nanoflakes filling the microcavities.

**Figure 3 nanomaterials-12-02772-f003:**
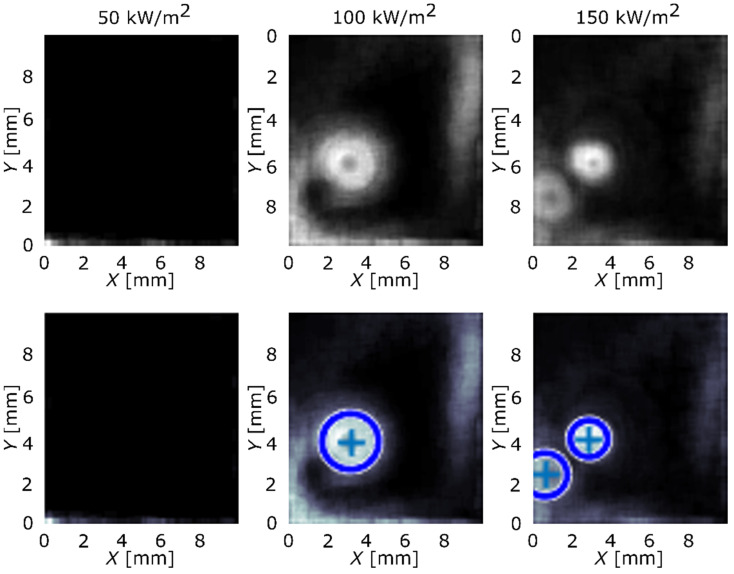
Active nucleation sites’ locations detected and located by the algorithm for surface REF.

**Figure 4 nanomaterials-12-02772-f004:**
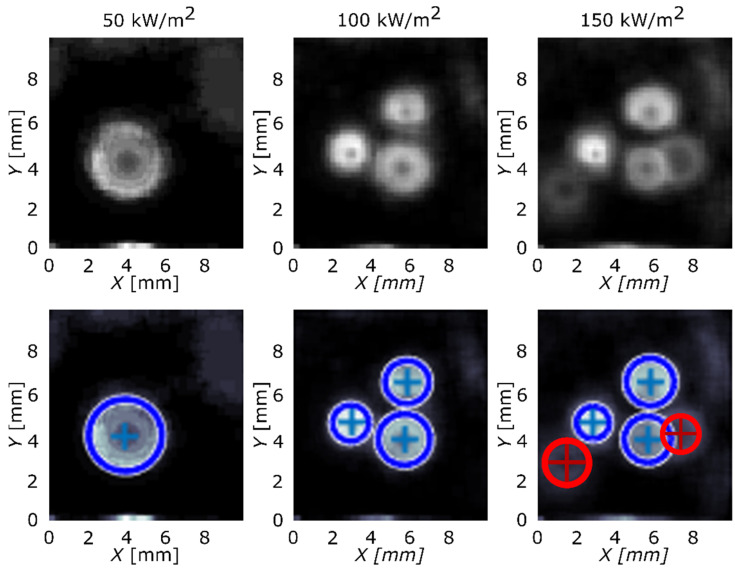
Active nucleation sites’ locations detected and located by the algorithm (blue circles) and manually (red circles) for surface GO2.

**Figure 5 nanomaterials-12-02772-f005:**
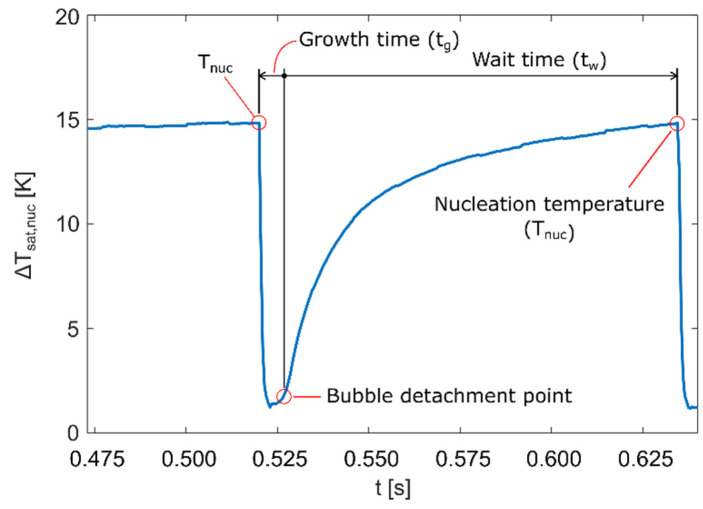
Wall superheat profile for a single nucleation measured at 50 KW/m^2^ on NS-REF sample.

**Figure 6 nanomaterials-12-02772-f006:**
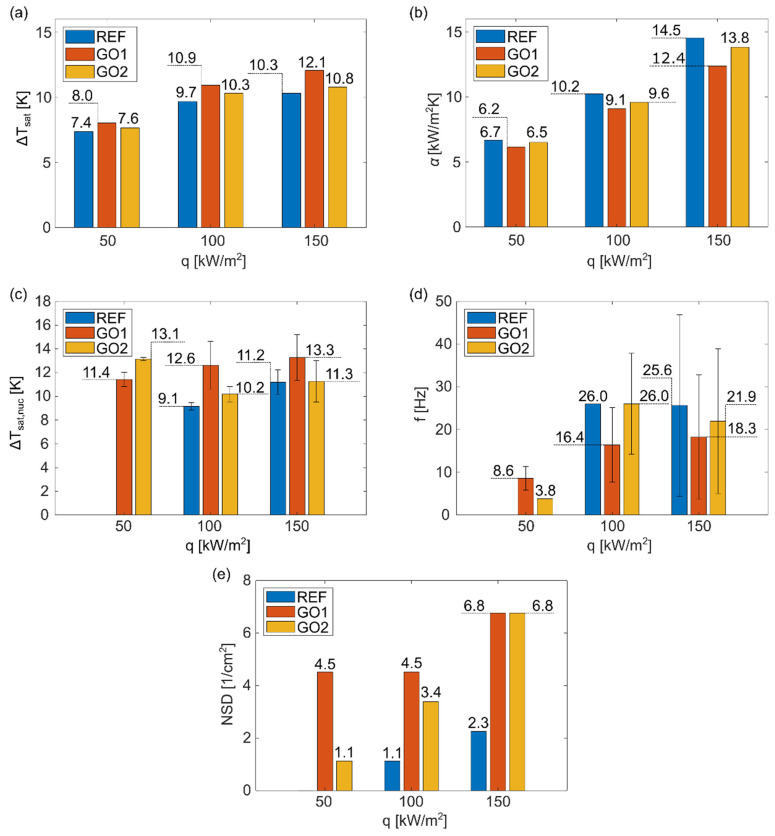
(**a**) Average wall superheat, (**b**) average heat transfer coefficient, (**c**) average nucleation wall superheat, (**d**) average nucleation frequency, and (**e**) nucleation-site density versus heat flux for samples REF, GO1, and GO2.

**Figure 7 nanomaterials-12-02772-f007:**
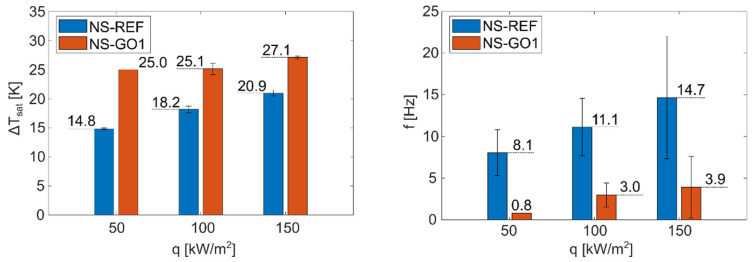
Average nucleation temperature superheat (**left**) and nucleation frequency (**right**) for NS-REF and NS-GO1 samples.

**Figure 8 nanomaterials-12-02772-f008:**
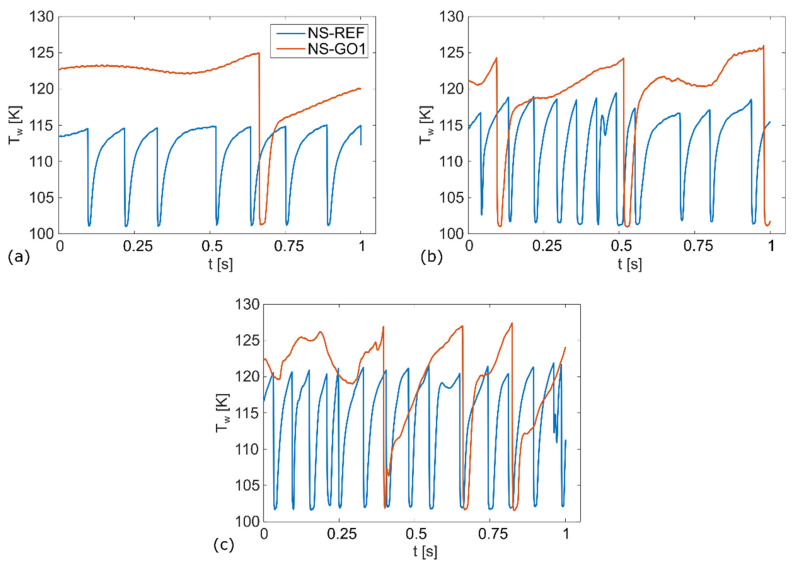
Wall-temperature profile for: (**a**) 50 kW/m^2^, (**b**) 100 kW/m^2^, (**c**) 150 kW/m^2^.

**Figure 9 nanomaterials-12-02772-f009:**
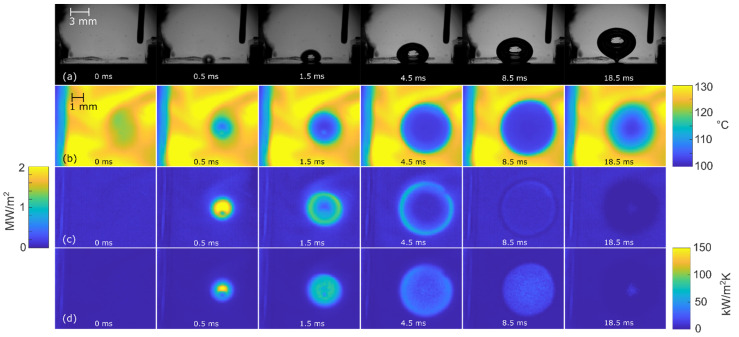
Bubble-growth process on surface NS-REF at 100 kW/m^2^: (**a**) high-speed camera video, (**b**) IR transient temperature field, (**c**) heat-flux distribution, and (**d**) HTC distribution.

**Figure 10 nanomaterials-12-02772-f010:**
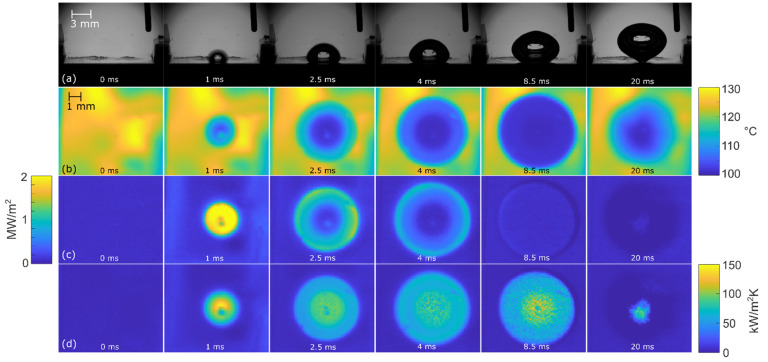
Bubble-growth process on surface NS-GO1 at 100 kW/m^2^: (**a**) high-speed camera video, (**b**) IR transient-temperature field, (**c**) heat-flux distribution, and (**d**) HTC distribution.

**Figure 11 nanomaterials-12-02772-f011:**
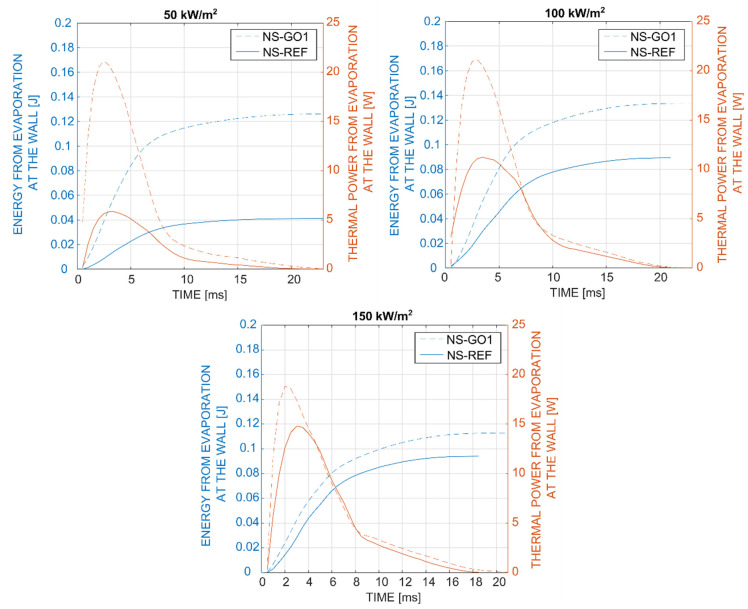
Comparison of the evaporation energy and thermal power from evaporation during the bubble growth on laser-textured nucleation site with (NS-GO1) and without (NS-REF) the graphene coating.

**Table 1 nanomaterials-12-02772-t001:** Static contact angle (*θ*) and surface roughness (*Ra*, *Rq*, and *Rz*) measurements.

Surface	REF	GO	GO1	GO2	NS-REF	NS-GO1
*θ* (°)	98	29	48	77	/	/
*Ra* (µm)	0.07	/	0.06	0.06	1.09	1.05
*Rq* (µm)	0.08	/	0.07	0.08	1.34	1.30
*Rz* (µm)	0.25	/	0.23	0.26	4.71	4.45

Note: Each reported result is an average of five repeated measurements.

## Data Availability

Data are available from the authors upon reasonable request.

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
