# Peer review of "Pool-Boiling Performance on Thin Metal Foils with Graphene-Oxide-Nanoflake Deposit"

_nanomaterials, 2022, doi:10.3390/nano12162772_

Round 1

Reviewer 1 Report

The manuscript titled «Pool boiling performance on thin metal foils with graphene oxide nanoflakes deposit» by Tadej Bregar, Matevž Vodopivec, Tim Pečnik, Matevž Zupančič and Iztok Golobič is devoted to the pool boiling performance of water on thin metal foils with graphene oxide deposition. The topic of the paper is relevant and in demand. The language of the paper is clear. However, as a reviewer I have several comments and questions to the authors. For now, my decision is major revision.

1.      In addition to the contact angle, the one of the most important surface parameters for boiling performance is the roughness. Please indicate the RMS and Ra values for the studied surfaces.

2.      Please add the corresponding boiling curves to the fig. 6.  Such data representation is more habitual.

3.      As I understood only heat fluxes up to 150 kW/m2 were studied in the paper. For the traditional boiling applications such thermal powers are too low, moreover the observed heat transfer enhancement for the fabricated surfaces is not significant. What the authors can say about larger heat fluxes and CHF values for the tested surfaces?

4.      As fig. 8 shows the nucleation frequencies for the individual nucleation site can vary significantly. What the authors can say about the statistical errors of its average values shown in fig. 6 and 7?

5.   It is necessary to add the nucleation site density values for the studied surfaces (NSD vs q plots).

6.     The most important note is that the paper lacks the description of the mechanisms of influence of surface modification on the boiling performance. E.g. using obtained data it is possible to compare the results on HTC with the mechanistic models (RPI and so on). Moreover, in the introduction the authors wrote that «The solution that combines ultra-thin properties and high thermal conductivity is utilization of graphene and graphene oxide». What the authors can say about the influence of thermal conductivity of the fabricated surfaces in HTC? Moreover the influence of the wettability should be described in more details. For now, the paper does not contain any significant results on the mechanisms of boiling enhancement and can only be considered as a description of the data.

Author Response

Please see the attached rebuttal letter.

Reviewer 2 Report

The  paper presents interesting finding on boiling phenomena observed over metal foams. Following comments are being shared for improving quality of the draft. 

Use of personal nouns (I, we etc.) in scientific writing should be avoided (has been used in abstract and methodology section, do correction in the whole text).

Methodology to be written in past tense (for example page 3 row 111, do corrections in the whole methodology section)

Figure 1 has not been discussed and referred in the text.

Figure 2 has not been discussed and referred in the text. 

List of reference is bit too long and some of the references are rarely discussed in the draft. It is suggested to skip unnecessary references. 

Author Response

Dear Reviewer,

thank you for your prompt revision of our manuscript. We have carefully evaluated and addressed your comments and provide a point-by-point response below.

Comment 1: Use of personal nouns (I, we etc.) in scientific writing should be avoided (has been used in abstract and methodology section, do correction in the whole text).

Response 1: Use of personal nouns was omitted throughout the text. In addition to that, proofreading was performed for the entire manuscript. We now resubmit the revised version with tracked changes.

Comment 2: Methodology to be written in past tense (for example page 3 row 111, do corrections in the whole methodology section).

Response 2: Thank you for this comment. Methodology is now written in the past tense (throughout the entire section).

Comment 3: Figure 1 has not been discussed and referred in the text.

Response 3: The reference to Figure 1 has been added to the first sentence in the section 2.1. "Pool boiling setup, schematically presented in Figure 1, encompassed a 50 × 50 × 50 mm3 boiling chamber, high-speed IR camera (FLIR SC6000), a high-speed video camera (Photron FASTCAM UX100), a light source (Opto Engineering LED LT2BC048036-W), and the data control and acquisition system based on CompactRIO real-time controller."

Comment 4: Figure 2 has not been discussed and referred in the text. 

Response 4: The reference to Figure 2 has been added in the last paragraph of section 2.2. "The results are shown in Table 1 and Figure 2, respectively."

Both the Figure 1 and Figure 2 were already discussed in the text in the original version of the manuscript, however, the references to those figures were missing. We thank the reviewer for this important remark.

Comment 5: List of reference is bit too long and some of the references are rarely discussed in the draft. It is suggested to skip unnecessary references. 

Response 5: The purpose of references cited in the introduction is to (i) highlight the potential problems of currently used coating-based technology in boiling heat transfer enhancement applications and to (ii) show potential usefulness and current work by utilising the graphene-based surface treatment. In places, where multiple references were used, we deleted some of the "less important ones". However, since the Reviewer 1 required some additional explanations behind the heater-thickness effects and heat flux partitioning, we were forced to include some extra references on this topics. Please see the revised version of the manuscript for the complete list of references.

Round 2

Reviewer 1 Report

The authors have done a great work and have answered all my quenstions and comments. The paper could be published in the present form.